# Deranged Myocardial Fatty Acid Metabolism in Heart Failure

**DOI:** 10.3390/ijms23020996

**Published:** 2022-01-17

**Authors:** Tsunehisa Yamamoto, Motoaki Sano

**Affiliations:** Department of Cardiology, Keio University School of Medicine, Tokyo 160-8582, Japan; tyamamoto19840625@gmail.com

**Keywords:** β-oxidation, triacylglyceride, PPAR, ERR, membrane fatty acid composition, SIRT1, SCD1, SFA, MUFA, ER stress

## Abstract

The heart requires fatty acids to maintain its activity. Various mechanisms regulate myocardial fatty acid metabolism, such as energy production using fatty acids as fuel, for which it is known that coordinated control of fatty acid uptake, β-oxidation, and mitochondrial oxidative phosphorylation steps are important for efficient adenosine triphosphate (ATP) production without unwanted side effects. The fatty acids taken up by cardiomyocytes are not only used as substrates for energy production but also for the synthesis of triglycerides and the replacement reaction of fatty acid chains in cell membrane phospholipids. Alterations in fatty acid metabolism affect the structure and function of the heart. Recently, breakthrough studies have focused on the key transcription factors that regulate fatty acid metabolism in cardiomyocytes and the signaling systems that modify their functions. In this article, we reviewed the latest research on the role of fatty acid metabolism in the pathogenesis of heart failure and provide an outlook on future challenges.

## 1. Introduction

To pump blood throughout the body, the heart must be beating and for it to continuously work, the heart must sustain the production of a large volume of adenosine triphosphate (ATP). In cardiomyocytes, ATP is produced by two reactions: mitochondrial oxidative phosphorylation and glycolysis. Mitochondrial oxidative phosphorylation usually accounts for 95% of ATP production, while glycolysis produces the remaining 5% [1,2]. The heart can use a variety of energy substrates such as fatty acids (FAs), lactate, glucose, ketone, and amino acids to maintain ATP production. Under normal conditions, 40–60% of ATP production is dependent on FAs [3,4]. FAs are taken up into cardiomyocytes via long-chain FA transporters (cluster of differentiation 36 (CD36), fatty acid transport protein (FATP), etc.,) on the cell membrane and converted to FA acyl-coenzyme A (CoA) by the long-chain FA acyl-CoA synthetase (ACSL1). A portion of FA acyl-CoA is taken up by mitochondria and used for ATP synthesis, and another portion is synthesized into triacylglycerides (TAG) in the smooth endoplasmic reticulum (ER) and packaged in lipid droplets for energy storage and synthesis of FA chains of phospholipids in the cell membranes.

Due to overeating or lack of exercise, excessive lipids are taken up by cardiomyocytes, which are non-fat tissues, leading to cardiomyocyte dysfunction and death. This phenomenon is known as lipotoxicity [5,6,7,8]. Not only the amount of FAs but also the quality of FAs needs to be considered. There is a wide variety of FAs with different numbers of carbons and different positions and numbers of double bonds. Different FAs and their metabolites have different effects on cardiomyocytes.

This review focused on mismatches between FA uptake, β-oxidation, oxidative phosphorylation, abnormalities in lipid droplet metabolism, and increased saturation of FAs in cell membrane phospholipids. We have summarized recent advances as “Recent advances in this area”, what we knew before as “What do we know in this area?”, the new concepts we discovered as “Membrane fatty acid composition and heart failure”, and the expected future treatment of heart failure as “Developing heart failure therapy by nutritional and pharmacological intervention”.

## 2. Fatty Acid Oxidation and Heart Failure

### 2.1. What Do We Know in This Area?

Analysis of obese Zucker diabetic fatty rats revealed that lipid accumulation in the heart is associated with cardiac dysfunction [9]. FA uptake exceeds its oxidation, resulting in lipid accumulation, which promotes myocardial injury [10]. For example, mice overexpressing lipoprotein lipase (LPL) or FATP1 showed increased FA uptake into the cardiomyocytes, leading to lipid accumulation and impaired contractile function [11,12]. In addition, mice overexpressing peroxisome proliferator-activated receptor α (PPARα), a fatty acid-activated nuclear receptor, showed increased expression of FA metabolic enzymes and FA transporters, leading to lipid accumulation in the cardiomyocytes and cardiac dysfunction [13]. Furthermore, contractile dysfunction in cardiac-specific PPARα-overexpressing mice was ameliorated by deleting the FA transporter CD36 or LPL [14,15]. Mitochondrial FA overload leads to the generation of reactive oxygen species (ROS), which causes myocardial damage.

Conversely, in hypertrophic hearts with reduced ejection fraction, FA metabolism is reduced and glucose utilization is increased [4,16,17,18]. Acetyl-CoA carboxylase 2 (ACC2) synthesizes malonyl-CoA, a potent endogenous inhibitor of carnitine palmitoyltransferase 1 (CPT1). Therefore, ACC2 is an indirect negative regulator of FA β-oxidation. Cardiac-specific ACC2-deficient mice showed sustained activation of FA metabolism, and these mice were resistant to angiotensin II-induced heart failure [19]. These results indicated that the “impaired FA metabolism” is itself responsible for heart failure. The decrease in FA metabolism causes an increase in glucose utilization, a phenomenon known as the Randle cycle [20]. The Randle cycle, also called the glucose-fatty acid cycle, is a metabolic process involving competition between glucose and FAs for substrates, i.e., the oxidation of FAs is enhanced and glucose utilization is decreased. The main mechanisms by which FA oxidation regulates glucose metabolism are (1) inhibition of hexokinase (HK), phosphofructokinase (PFK), and glucose transporter (GLUT) by citrate produced by FA metabolism, (2) inhibition of HK by FA-acyl CoA produced by FA metabolism, (3) activation of protein kinase C (PKC) by diacylglycerol (DAG), and (4) PKC inhibits insulin signaling by phosphorylating the insulin receptor substrate (IRS) [21,22,23,24,25]. Decreased glucose metabolism is observed in PPARα overexpression models [13], and increased glucose metabolism is observed in decreased FA oxidation models with the knockout of krüppel-like factor (KLF) 5 [26], LPL [27], CD36 [28], and fatty-acid binding protein (FABP) 4/5 [29]. In the diabetic heart, impaired glucose metabolism is observed mainly due to impaired insulin signaling caused by IRS inhibition [25].

Activation of glucose metabolism in the process of cardiac hypertrophy can be viewed as an active change rather than just a secondary change as a result of reduced FA oxidation. Activation of the aerobic glycolytic system as a metabolic change necessary to maintain cell proliferation in cancer cells is known as the Warburg effect [30]. Recently, the significance of the Warburg effect is suggested to be that it provides synthetic materials for biomolecules (nucleic acids and nicotinamide adenine dinucleotide phosphate (NADPH)) as by-products of the glycolytic system. Among the two isoforms of pyruvate kinase M (PKM1 and PKM2) that are generated by selective splicing, the selective expression of PKM2 has been reported to contribute significantly to the Warburg effect in cancer cells. This selective expression of PKM2 in cancer cells causes low glucose-to-pyruvate flux and restricted carbon source entry into the tricarboxylic acid (TCA) cycle. Although uncoupling between glycolysis and glucose oxidation is also observed in failing myocardium, PKM2 expression is higher in failing myocardium than in normal myocardium [31]. Besides glycolysis, glutaminolysis is another main pillar for energy production in tumor cells, however, this is not the case in cardiac muscle. Cardiomyocytes, whether healthy or hypertrophic, utilize very little glutamine [32,33].

GLUT family proteins are the major players in glucose transport in the heart. The most abundant glucose transporters in the heart are GLUT1 and GLUT4. GLUT1 is mainly localized to the plasma membrane and responsible for most of the basal myocardial glucose uptake. On the other hand, GLUT4 is localized in intracellular vesicles at rest and is translocated to the plasma membrane upon insulin stimulation. In the hypertrophied heart, GLUT1 expression and basal glucose uptake are increased, but GLUT4 expression levels and insulin-mediated glucose uptake are decreased. Similar observations have been made in patients with compensatory hypertrophy [34]. In hypertrophic and failing hearts, the glycolytic rate is relatively increased compared to that in the oxidation of pyruvate derived from glucose (i.e., glucose oxidation). Whether glucose oxidation is increased or decreased in hypertrophic or failing hearts is not well understood. There are many reports on decreased glucose oxidation in failing hearts [35,36,37], albeit there are also reports on increased glucose oxidation [19,38]. However, it can be stated that the degree of activation of glucose oxidation is lower than that of the glycolytic system. The mechanism of how activation of the glycolytic system contributes to cardiac remodeling is explained by the by-products of nonoxidative glucose metabolism derived from glycolysis, such as the pentose phosphate pathway, hexosamine biosynthetic pathway, galactosamine synthesis, and mannose biosynthetic pathway [39]. Previously, we have shown that the activation of pyruvate dehydrogenase (PDH) by dichloroacetate (DCA) enhances pyruvate uptake and suppresses cardiac hypertrophy and heart failure induced by pressure-overload. Furthermore, the results of a fluxome analysis using ^13^C-labeled glucose showed that the activation of PDH by DCA also affected the histone acetylation and gene regulation by increasing the acetyl-CoA pool in the cardiomyocytes [40]. These results suggest that uncoupling of the glycolysis and glucose oxidation is involved in cardiac hypertrophy and impaired cardiac function and that increasing the efficiency of pyruvate uptake into the mitochondria is a promising strategy for the treatment of cardiac hypertrophy and heart failure (Figure 1).

The PPAR family is a nuclear receptor that controls FA uptake and β-oxidation [41]. PPARα plays a central role in FA metabolism in the heart, and cardiac-specific PPARα-deficient mice exhibit impaired FA metabolism and heart failure [42,43,44]. On the other hand, overexpression of PPARα also causes cardiac dysfunction due to a mismatch between FA uptake and utilization [13]. The estrogen receptor-related receptor (ERR) family of nuclear receptors is known to regulate a wide range of gene expression in cardiomyocytes, including β-oxidation of FA, oxidative phosphorylation, and contractile proteins [45]. Cardiac-specific ERR-deficient mice exhibit mitochondrial dysfunction, leading to heart failure [46]. On the other hand, it has been shown that overexpression of ERRγ, a member of the ERR family, in cardiomyocytes causes cardiac hypertrophy via the GATA-binding protein 4 (GATA4) [47,48].

As described above, to efficiently produce ATP and maintain ideal cardiac function, the processes of FA uptake, β-oxidation, and mitochondrial oxidative phosphorylation must work together smoothly and in balance. Even if the capacity of any of these processes is increased, if the downstream processes are not able to accept it, it will lead to the failure of cardiac function. The nuclear receptors PPAR and ERR are important transcription factors that regulate FA uptake, β-oxidation, and mitochondrial oxidative phosphorylation. However, both overexpression and knockout of each transcription factor cause heart failure. Both PPAR and ERR need to be fine-tuned within the physiological range. As a next step, several reports on molecules involved in metabolic control by PPAR and ERR were presented. Among the transcription factors of the KLF family, KLF15 and KLF5, cooperate with PPARα to regulate FA metabolism genes [26,49]. In mice with cardiomyocyte-specific deletion of KLF5 or 15, reduced expression of FA metabolism genes was observed [26,50,51]. Another KLF family transcription factor, KLF4, regulates oxidative phosphorylation in mitochondria in concert with ERR, and cardiomyocyte-specific KLF4-deficient mice develop heart failure [52].

PPARγ coactivator-1 (PGC-1) family of transcription coactivators are highly responsive to a variety of environmental cues, from temperature to nutritional status to physical activity. In the normal heart, PGC-1α coordinately regulate mitochondrial oxidative metabolism, via coactivating PPAR, ERR, and nuclear respiratory factor 1/2 (NRF1/2) [53]. Overexpression of PGC-1α in the mouse heart under the control of the cardiac muscle-specific alpha-myosin heavy chain (α-MHC) promoter leads to progressive mitochondrial biogenesis, to the extent that the myofibrillar apparatus is replaced by the mitochondrial matrix [54,55]. In contrast, PGC-1α-deficient mice have reduced mitochondrial number and respiratory capacity, and their hearts are vulnerable to pressure-overload [56,57]. The role of co-repressors and co-activators of PPARs and ERRs should also be highlighted, such as chicken ovalbumin (OVA) upstream promoter transcription factor 2 (COUP-TFII, also known as NR2F2) and receptor-interacting protein 140 (RIP140) are known to suppress the activity of PPAR and ERR. Cardiac-specific overexpression models of these factors have been reported to cause heart failure due to the suppression of PPAR and ERR and disruption of mitochondrial function [58,59].

It is unclear why hypertrophic or failing myocardium has “impaired FA oxidation and enhanced glucose metabolism”. In the failing myocardium, transcriptional factors, such as PPAR/ERR, PGC-1, and KLF, become deactivated and FA oxidation is reduced. Is there a primary decline in FA oxidation, causing a “loss of Randle cycle”, which raises glucose metabolism? However, even in that case, it does not explain why uncoupling occurs. In the loss of the Randle cycle, both glycolysis and glucose oxidation should increase. There may also be a factor, such as pressure-overload, which acts as a primary driver to actively increase glucose utilization. Further studies are needed to answer this question.

### 2.2. Recent Advances in This Area

In 2020 and 2021, a series of findings referring to the role of mitochondrial pyruvate carriers (MPCs) in heart failure were reported [60,61,62,63]. MPCs are downregulated in human and mouse failing hearts, and pyruvate oxidation is reduced. The authors observed cardiac function in mice deficient in cardiac-specific MPC1 and/or MPC2 to determine whether reduced expression of MPCs is related to the development of heart failure. The results revealed that loss of MPCs caused uncoupling of glycolysis and glucose oxidation, causing cardiac hypertrophy and heart failure. Conversely, overexpression of MPC1 had a protective effect on cardiac hypertrophy induced by pressure-overload surgery [60]. Of note, cardiac dysfunction in MPC-deficient hearts can be rescued by a high-fat diet or a ketogenic diet [61,62].

Although aspartate has been previously reported to increase in hypertrophic and failing hearts [28,64,65], Ritterhoff et al. found that glucose supplies carbon to aspartate, which supplies the nitrogen necessary for nucleic acid synthesis during cardiac hypertrophy [33]. Authors also revealed that carbon from pyruvate was supplied to aspartate not only through its conversion to acetyl-CoA by PDH but also through its conversion to oxaloacetate by pyruvate carboxylase (PC). These results suggest that pyruvate derived from glucose contributes to aspartate formation through various influx pathways via PC in addition to MPC, although uncoupling of glycolysis and glucose oxidation occurs. Additionally, authors demonstrated that increasing FA oxidation by ACC2 knockdown inhibited the generation of by-products from the pentose phosphate pathway and aspartate and consequently inhibited cardiac hypertrophy.

Restoring FA metabolism is effective not only for pressure-overload-induced contractile dysfunction but also for high-fat diet (HFD)-induced contractile dysfunction. HFD loading decreases the expression of parkin, a regulator of mitochondrial autophagy (mitophagy). The heart with reduced parkin expression exhibited mitochondrial dysfunction, cardiac hypertrophy, and contractile dysfunction. However, in enhanced FA metabolism heart by cardiac-specific ACC2 knockdown, there was no HFD-induced reduction in parkin, and cardiac hypertrophy and contractile dysfunction were mild [66]. Although PPARα activity is known to be increased in obesity- and diabetes-induced cardiomyopathy, the target genes of PPARα are biased in mice fed with HFD. In the hearts of HFD-fed mice, phosphorylation of Ser280 of PPARα by glycogen synthase kinase 3α (GSK3α) induces a biased response that shifts the gene target of PPARα to FA uptake rather than β-oxidation. On top of that, it has also been shown that fenofibrate inhibits GSK3α activation under HFD, thereby suppressing Ser280 phosphorylation of PPARα and the associated biased PPARα target gene expression [67].

## 3. Triacylglyceride Dynamics and Heart Failure

### 3.1. What Do We Know in This Area?

Myocardium prefers FA as a substrate for ATP production, albeit the myocardium cannot produce FA directly. There are three ways to obtain FA: (1) Uptake of FA released from adipocytes into blood; (2) uptake of FA into cardiomyocytes through lipolysis of triacylglyceride (TAG)-lipoprotein by LPL on vascular endothelial cells; and (3) production of FA through lipolysis of endogenous TAG in the myocardium. In healthy myocardium, FA taken in from outside is more actively used as a substrate for ATP production than FA obtained by lipolysis of endogenous TAG in the myocardium [68]. In diabetic cardiomyopathy, the turnover rate of the endogenous TAG pool becomes faster and the rate of endogenous FA oxidation increases [69]. LPL, important for the uptake of FA from the blood into the myocardium, is produced and secreted by cardiomyocytes. LPL is excised by heparanase (Hpa) generated by endothelial cells and further translocated to the surface of vascular endothelial cells by the action of glycosylphosphatidylinositol-anchored high-density lipoprotein-binding protein 1 (GPIHBP1) [70]. In the LPL overexpression model, excessive FA supply to the cells causes myopathy, cardiac enlargement, contractile dysfunction, and glucose intolerance [11,71,72]. Additionally, cardiac-specific LPL-deficient mice elevate blood triglycerides, decrease FA metabolism, decrease expression of PPARα-target genes, and cause contractile dysfunction [27,73]. When vascular endothelial cells are cultured under hyperglycemic conditions, Hpa secretion is enhanced [74]. Patients with type 2 diabetes are reported to have high levels of Hpa in their blood [75]. Additionally, the expression of GPIHBP1 in vascular endothelial cells is also upregulated in the hearts of diabetic mice [76]. These results indicate that hyperglycemia increases LPL expression in the vascular endothelium, which causes increased FA uptake and a mismatch between uptake and utilization in cardiomyocytes. This mismatch leads to the formation of lipid droplets and the accumulation of DAG and ceramide as observed in diabetic cardiomyopathy.

The TAG pool in lipid droplets is not physiologically inert but is dynamically metabolized and plays an important role in the regulation of FA metabolism in cardiomyocytes [77]. The changes in TAG content and its contribution to metabolism were first recognized by O’ Donnell et al. They noted that TAG volume and TAG turnover rate decreased in pressure-overloaded failing hearts. The amount of TAG was also reduced in human failing heart samples, although it is not possible to measure TAG turnover in human samples [68]. Furthermore, decreased myocardial TAG content in heart failure patients was also associated with elevated levels of toxic lipid intermediates such as DAG and ceramide [78]. In failing hearts, the activity of PPARα is reduced, and the expression of FA metabolism genes is decreased [17,79]. It should be noted that the signal of PPARα does not depend on the static content of TAG but on the release of fatty acyl groups from TAG as ligands for PPARα activation, that is, on the dynamics of TAG turnover. Lipolysis of intracellular TAG by adipose triglyceride lipase (ATGL) generates lipid ligands for PPAR activation. Deletion of ATGL reduces the mRNA levels of PPAR-target genes, causing impaired mitochondrial substrate oxidation and respiration, excessive lipid accumulation, heart failure, and fatal cardiomyopathy [80]. In the loss of function model of diacylglycerol acyltransferase (DGAT) 1 and 2, responsible for TAG production, TAG content is reduced, TAG dynamics is disrupted, and the expression of PPAR-target genes is reduced [81]. In a systemic knockout model of Perilipin5, a lipid droplet-binding protein, TAG content is reduced and chronic mitochondrial FA overload leads to increased ROS and contractile dysfunction [82]. Conversely, cardiac-specific DGAT1 overexpression models exhibit increased TAG content and TAG dynamics, which are protective against ischemia-reperfusion injury [83]. Shortage of the TAG pool during cardiac decompensation would be expected to cause a further decrease in PPAR activity and exacerbation of cardiac function.

FAs are classified based on the number of double bonds. Saturated fatty acids (SFA) have no double bonds, monounsaturated fatty acids (MUFA) have one double bond, and polyunsaturated fatty acids (PUFA) have two or more double bonds. Palmitate, a representative SFA, causes cell dysfunction, but oleate, a representative MUFA, does not cause dysfunction and is known to reduce the toxicity of palmitate. Several studies have reported the mechanisms involved in the differential effects of SFA and MUFA on cell function [84,85,86,87,88]. One of them is the difference in the effects of palmitate and oleate on the dynamics of TAG turnover [89]. Using the Langendorff perfusion cardiac experimental method, differences in TAG kinetics and cardiac function were observed when palmitate or oleate was used as the energy substrate in hypertrophied hearts. When perfused with palmitate, the TAG content, TAG kinetics, and expression of PPARα target genes were lower in the hypertrophied hearts than in the normal hearts, and contractile function was reduced. On the other hand, when perfused with oleate, the TAG content, TAG kinetics, and expression of PPARα target genes were similar to those of normal hearts, and contractile function was maintained. This result supports the fact that oleate can activate TAG turnover more than palmitate and that TAG turnover is linked to the maintenance of cardiac function by supplying ligands for PPARα (Figure 1).

### 3.2. Recent Advances in This Area

FAs need to be converted to acyl-CoA by ACSL1 for β-oxidation or TAG formation. Acyl-CoA levels are reduced in failing hearts in humans and hypertrophic hearts in mice [90]. Therefore, an experiment was conducted to test the hypothesis that maintaining acyl-CoA levels would improve heart failure. In cardiac-specific ACSL1 overexpressing mouse heart, the amount of acyl-CoA was increased, and TAG turnover was not decreased after pressure-overload surgery, and cardiac function was maintained [90]. Furthermore, improvement in the amount of acyl-CoA has been observed in human myocardial tissue subjected to mechanical unloading with a left ventricular assist device (LVAD) [90]. Interestingly, the ceramide profile was also significantly altered in ACSL1-overexpressing hearts. Several reports have identified C16, C24, C24:1 ceramides as toxic ceramide that are associated with worsening cardiac function, while C20 and C22 ceramides are cytoprotective ceramides [89,91,92]. These toxic ceramides were found to be increased in the hearts of hypertrophic mice. However, in ACSL1-overexpressed hearts, an increase in toxic ceramides was not observed; instead, an increase in cytoprotective ceramide was observed after pressure-overload surgery. Increasing the amount of acyl-CoA via activation of ACSL1 in failing hearts may improve TAG turnover and ceramide profile (decrease in toxic ceramides and increase in cytoprotective ceramides) (Figure 1). However, it is still unclear how TAG dynamics and ceramide synthesis are related and regulated, and this is a subject for future research.

## 4. Membrane Fatty Acid Composition and Heart Failure-New Concept Targeting Lipid Metabolism in Heart Failure

Palmitate (C16:0 SFA) is solid at room temperature, and oleate (C18:1 MUFA) is liquid at room temperature. Because of this difference in physical properties, it is easy to understand why palmitate and oleate have opposing effects on cellular homeostasis [89].

In Chinese hamster ovary (CHO) cells, exogenously given palmitate is first rapidly incorporated into the phospholipid component of the ER membrane, resulting in the loss of ER structure and function [93]. Stearoyl-CoA desaturase 1 (SCD1), an enzyme that catalyzes the rate-limiting conversion of SFA to MUFA and plays a protective role against palmitate-induced toxicity, is expressed in the ER membrane [94,95,96]. These observations suggest that changes in the FA composition of membrane phospholipids in the ER membrane may play a role in SFA-induced toxicity.

We compared the hearts of mice fed an SFA-rich HFD with those fed a MUFA-rich HFD [97]. There were no differences in FA uptake, β-oxidation, TAG turnover, or lipid intermediate content such as those of DAG and ceramide between the SFA-rich and MUFA-rich HFD-fed hearts. However, in hearts fed SFA-rich HFD, supersaturation of FA chains of membrane phospholipids (increased SFA/MUFA ratio), activation of the protein kinase R-like ER kinase (PERK), inositol-requiring enzyme 1 (IRE1), and decreased left ventricular diastolic function were found. This means that even in cardiomyocytes, which use FAs as their main energy source, the FA composition of cell membrane phospholipids can be affected by FAs taken into the cell from outside, as in CHO cells [97]. In addition, SCD1 expression was also suppressed by SFA-rich HFD.

Sirtuin1 (SIRT1), an NAD^+^-dependent deacetylase, has been identified as an upstream regulator of SCD1 [98]. SIRT1 is an anti-aging molecule that mediates the favorable effects of calorie restriction and exercise [99]. The expression of SIRT1 was decreased by SFA treatment. In the heart, SIRT1 regulates the expression of SCD1 via the nuclear receptor liver X receptor (LXR). When cardiac-specific SIRT1-deficient mice were fed SFA-rich HFD, the very low expression of SCD1 accelerated the saturation of membrane phospholipids (increase in the SFA/MUFA ratio) and exacerbated ER stress and left ventricular diastolic dysfunction compared to that in control mice. Furthermore, the treatment of mice with nicotinamide mononucleotide (NMN), an NAD^+^ precursor, increased SIRT1 activity and SCD1 expression, and ameliorated SFA-rich HFD-induced increase in the SFA/MUFA ratio, ER stress, and left ventricular diastolic dysfunction [98]. These results suggest that SFA reduces the expression and activity of SIRT1 in the myocardium, thereby decreasing SCD1 expression and accelerating the saturation of membrane phospholipids. Thus, to avoid SFA-induced diastolic dysfunction, it is important to prevent the saturation of membrane phospholipids in cardiomyocytes. Activation of SIRT1/SCD1 signaling by exercise, caloric restriction, or NMN administration may be a good therapeutic option.

Given that the FA composition of membrane phospholipids is directly influenced by exogenous FA ingested from the diet, a therapeutic effect could be expected by making the FAs in the diet MUFA-centered. To test this hypothesis, we switched HFD from SFA-rich HFD to MUFA-rich HFD and analyzed HFD-fed mouse hearts. Compared to those of whole-time SFA-rich HFD-fed hearts, the hearts switched to MUFA-rich HFD revealed better SIRT1/SCD1 signal, membrane FA saturation, and ER stress induction. Left ventricular diastolic dysfunction was also less severe in the hearts switched to MUFA-rich HFD compared to the whole-time SFA-rich HFD-fed heart. Restoration of SIRT1/SCD1 signaling is important as part of the effect of switching to MUFA-rich HFD. Furthermore, the effect of switching to MUFA-rich HFD was also confirmed in cardiac-specific SIRT1-deficient hearts, suggesting that MUFA can directly modify membrane FA composition independently of SIRT1/SCD1 signaling [100]. Increasing the amount of MUFA in the diet or switching from SFA to MUFA may have beneficial effects even in hearts with low SIRT1 activity, such as aged hearts [101].

Thus, it is strongly suggested that ER stress owing to saturation of cell membrane phospholipids contributes to the formation of left ventricular diastolic dysfunction associated with overeating and lack of exercise (Figure 2). Although saturation of membrane phospholipids has been reported to cause cardiomyocyte death in an X-box binding protein 1 (XBP1)-independent manner through the IRE1 pathway [102], more detailed studies will be required to clarify the mechanism of SFA-induced left ventricular diastolic dysfunction.

## 5. Developing Heart Failure Therapy by Nutritional and Pharmacological Intervention

### 5.1. Statin and Fibrate Therapy

Currently, there are only a few ways to intervene in myocardial FA metabolism. In the late 1990s, the structure and function of PPARα were elucidated, and it became clear that one of the targets of action of statins and fibrates was activation of PPARα [103,104]. Statin, as an HMG-CoA reductase inhibitor, is known to lower cholesterol and inhibit cardiovascular events, and PPARγ activation in macrophages has also been reported to be involved in the anti-atherosclerotic effect of statin [105]. Statins have the potential to prevent cardiac hypertrophy through multifaceted activation of PPARα and PPARγ and by inhibiting inflammation and fibrosis [106,107]. Fibrate has been reported to inhibit cardiac hypertrophy and heart failure through suppression of PI3K/Akt/mTOR signaling and regulation of HMGB1 expression, in addition to activation of PPARα [104]. Kawamoto et al. revealed that activation of PPARα after pressure-overload surgery increased PPARα-target genes related to FA metabolism and preserved cardiac contractile function. This suggests that in the early stages of heart failure, the role of PPARα activation may be to restore FA oxidation, thereby, restoring energy production and cardiac function [108]. Pemafibrate was developed based on the concept of increasing the selectivity of fibrates for PPARα and reducing the burden on the liver and kidneys. The PROMINENT trial, a large-scale clinical trial of pemafibrate, is currently underway worldwide [109], and we look forward to the results to know if it has a preventive effect on heart failure.

### 5.2. Ketone Body Supplementation and SGLT2 Inhibitors

Ketone bodies (acetoacetate, β-hydroxybutyrate, and acetone) are a general term for the metabolites of lipolysis. In patients with heart failure with reduced ejection fraction (HFrEF), blood ketone concentrations are elevated and there is an increased reliance on ketone bodies as a substrate for myocardial ATP production [4,110]. Arima et al. revealed that in the hearts of Hmgcs2 knockout mice, which are unable to produce ketone bodies, acetyl-CoA accumulates intracellularly, causing increased acetylation of mitochondrial proteins and decreased mitochondrial function [111]. Yan Deng et al. first generated a three-hit heart failure with preserved ejection fraction (HFpEF) mouse model combining aging, long-term HFD, and desoxycorticosterone pivalate (DOCP). In this HFpEF heart, they observed impaired mitochondrial function secondary to increased acetyl-CoA pool and acetylation of mitochondrial proteins. β-hydroxybutyrate can reduce the acetyl CoA pool, improve mitochondrial function and cardiac function in this three-hit HFpEF model [112]. The way to elevate blood ketone body levels includes ketogenic diet and ketone supplements (1.3-butanediol, medium-chain triglyceride, ketone salts, and ketone ester). Sodium-glucose cotransporter 2 (SGLT2) inhibitors may be also useful in maintaining high blood levels of ketone bodies. SGLT2 inhibitors, marketed as hyperglycemic agents, have been proven to be effective in the treatment of heart failure with or without diabetes [113,114,115]. The mechanism of action varies widely [116], however, one possible mechanism is the cardioprotective effects of increased ketone body production associated with increased glucagon/insulin ratio [117]. Ketone bodies have been observed not only to exert anti-cardiac effects when utilized as an energy substrate but also to act in an organoprotective manner through signaling and epigenetic regulation. It is well-known that mammalian target of rapamycin (mTOR) signaling is involved in the pathogenesis of cardiac hypertrophy and diabetic nephropathy. Ketone bodies have a protective effect on cardiac hypertrophy and diabetic nephropathy by inhibiting mTOR signaling [118,119]. In addition, ketone body inhibits class I histone deacetylases (HDACs) and activate Notch signaling in intestinal epithelial cells, thereby promoting stem cell self-renewal [120]. Furthermore, the ketone body-mediated inhibition of HDACs increased histone acetylation at the Foxo3a and Mt2 promotors and enhances the antioxidant stress response [121].

### 5.3. Short-to Medium-Chain Fatty Acid Supplementation

Short- to medium-chain FAs can pass through the mitochondrial inner membrane without being converted to acylcarnitines. Ischemic cardiac muscle has been reported to actively take up short-chain FAs into the mitochondria [122]. A diet rich in short- to medium-chain FAs can reportedly improve heart failure induced by impaired FA metabolism in the loss of function model of CD36 [28], FABP4/5 [29], and KLF15 [51]. A medium-chain FA-rich diet reportedly improves FA metabolism and cardiac function in ATGL knockout heart, which do not receive TAG-derived long-chain FA [123].

### 5.4. MUFA and PUFA Supplementation

Zhuang et al. conducted a large cohort study that examined the relationship between lipid intake and mortality. Using carbohydrate-only energy intake as the reference (hazard ratio 1.0), the hazard ratio for all-cause mortality with increased SFA intake was 1.09 and increased MUFA intake was 0.94. This large cohort study also examined the effect of replacing dietary SFA with unsaturated FA on the all-cause mortality hazard ratio. Replacing 5% of the calories from SFA with MUFA changed the hazard ratio for total mortality to 0.82 and the hazard ratio for cardiovascular disease mortality to 0.85. The results of this study indicate that the type of dietary fat can significantly influence the incidence of cardiovascular events and MUFA intake can have a protective effect [124]. Additionally, the results of an interview study of patients with HFpEF reported that those who consumed more unsaturated FA (MUFA and PUFA) daily had improved left ventricular diastolic function and exercise tolerance [125]. There is also a report of improved left ventricular diastolic function and exercise tolerance in patients with HFpEF following 12 weeks of active supplementation with unsaturated FA [126]. The effect of PUFA on the suppression of cardiovascular events has not yet been conclusively established [127,128,129,130,131], however, PUFA metabolites, 18-HEPE, have been reported to have anti-inflammatory and anti-fibrotic effects [132]. It would be important to consider not only the total amount of lipids in the diet but also the quality (short- to medium-chain FA vs. long-chain FA and saturated FA vs. unsaturated FA).

### 5.5. Boosting NAD^+^ Level by Supplementation

Lower NAD^+^ levels or lower NAD^+^/NADH ratios have been observed in failing hearts [133,134,135]. NAD exists in oxidized (NAD^+^) and reduced (NADH) forms and acts as a major coenzyme for electron transfer in glycolysis, the TCA cycle, and oxidative phosphorylation. NAD^+^ is also important as a substrate for poly [ADP-ribose] polymerases (PARPs), Sirtuin, and CD38 and plays a role in translational modification, DNA damage repair, and gene transcription [39]. Additionally, the efficacy of NAD supplementation on heart failure has been confirmed in mice and humans [134,135,136,137,138,139].

## 6. Conclusions

Lipids taken up by the myocardium follow various metabolic pathways and play an important role in ATP production and maintenance of pump function. Lipid metabolism also regulates glucose metabolism, but the regulatory mechanisms remain unclear. In failing hearts, the activity of transcription factor networks such as PGC-1/KLF/PPAR/ERR is impaired. The transcription factor network is regulated by non-coding RNAs, DNA methylation, histone modifications, and higher-order chromatin structures, but further detailed analysis of how these are altered by pathological conditions such as obesity, diabetes, heart failure, and aging is needed. In addition, there are still many points that need to be clarified, such as the interaction between metabolites resulting from lipid metabolism and all networks, as well as the linkage with multiple organs. However, step by step, we continue to create new therapeutic interventional plans based on research results. We can already see the light to overcome cardiovascular diseases.

## Figures and Tables

**Figure 1 ijms-23-00996-f001:**
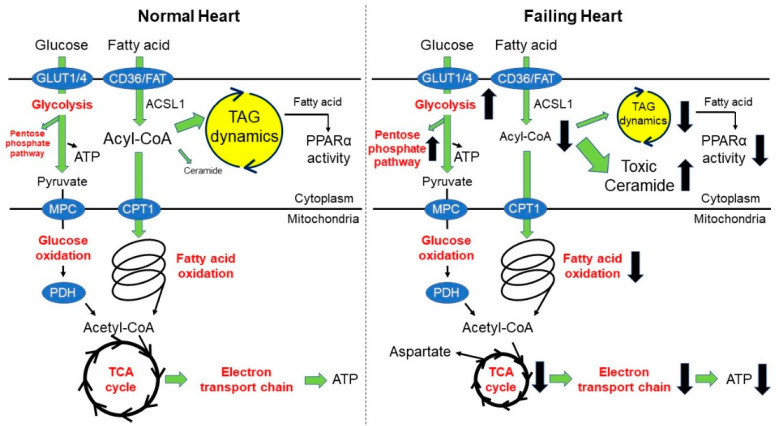
Energy metabolism in the normal heart and failing heart. In the normal heart, glucose is transported into the cardiomyocyte by GLUT1 or GLUT4. Further, glucose undergoes glycolysis to produce pyruvate, which is transported to the mitochondria by MPC. Transported pyruvate is converted to acetyl CoA by PDH. Fatty acids are transported to the cardiomyocyte by CD36 and FAT. Fatty acids are converted to fatty acid acyl-CoA by ACSL1. The acyl-CoA is transferred to carnitine by CPT1 and transported into mitochondria to undergo fatty acid oxidation producing acetyl-CoA. The acyl-CoA is also used for producing TAG. Dynamic turnover of TAG release fatty acids as a ligand for PPARα. In the failing heart, alterations in glycolysis, glucose oxidation, fatty acid oxidation, TAG dynamics, TCA cycle, and electron transport chain are observed. Uncoupling of glycolysis and glucose oxidation produce by-products for anabolic reaction (i.e., pentose phosphate pathway, aspartate). A black arrow facing upwards indicates an increase and downwards indicates a decrease. ACSL1, Acyl-CoA synthetase long-chain family member 1; ATP, adenosine triphosphate; CPT1, carnitine palmitoyltransferase I; FAT, fatty acid transporter; GLUT, glucose transporter; MPC, mitochondrial pyruvate carrier; PDH, pyruvate dehydrogenase; PPARα: peroxisome proliferator-activated receptor alpha; TCA cycle, tricarboxylic acid cycle; and TAG, triacylglyceride.

**Figure 2 ijms-23-00996-f002:**
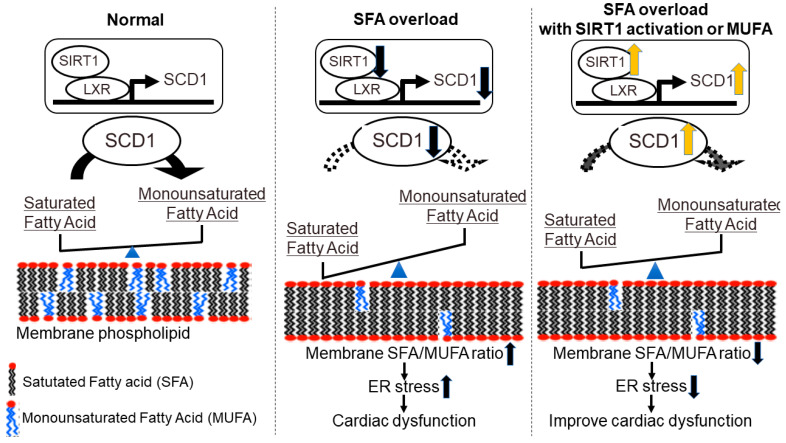
The relationship between cardiomyocyte membrane fatty acid composition and contractile dysfunction. Cardiomyocyte membrane fatty acid composition, the balance of SFA and MUFA, are well maintained by the SIRT1-LXR-SCD1 axis. SFA overload decreases the activity of the SIRT1-LXR-SCD1 axis, which leads to a high ratio of membrane SFA/MUFA. An increase in membrane SFA/MUFA ratio triggers ER stress and relates to cardiac contractile dysfunction, mainly diastolic dysfunction. Intervention with either SIRT1 activation or MUFA supplementation during SFA overload improves membrane SFA/MUFA ratio, ER stress, and contractile dysfunction. A black arrow facing upwards indicates an increase and downwards indicates a decrease. ER, endoplasmic reticulum; LXR, liver X receptor; MUFA, monounsaturated fatty acid; SCD1, stearoyl-CoA desaturase I; SFA, saturated fatty acid; and SIRT1, sirtuin1.

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
