# Peer review of "Deranged Myocardial Fatty Acid Metabolism in Heart Failure"

_ijms, 2022, doi:10.3390/ijms23020996_

Round 1
Reviewer 1 Report
The manuscript by Yamamoto and Sano represents a narrative review about the role of fatty acid metabolism in the pathogenesis of heart failure and future challenges. However, the review is not novel there is no mention of how the cited references were selected and the literature review is incomplete. The authors should explain what is the novelty of this review. There are several new references, but if the aim of this paper is update of the literature data, most of the references should be from the last several years. Therefore, an update of the references with a clear novelty should be done.
Further, the authors presented 2 nice figures on the mechanisms in normal and cardiac disfunction, but without Figure legends, which should be added. I also suggest to add a chapter on the effects of nutritional intervention and the statin/fibrate therapy on the FA metabolism in membranes.
After these corrections manuscript can be reconsidered for publication since a comprehensive up-to-date literature review of the FA metabolism in heart failure is of a broad interest.
Reviewer 2 Report
The review is a timely summary of a very important topic. Some questions and comments are offered for consideration.
Fatty acid oxidation and heart failure
- The evidence presented in this section seems to indicate that metabolic remodeling of impaired FA metabolism and increased glucose utilization responsible for heart failure; but perhaps this is simply association? More explanation is needed for the role played by uncoupling of the glycolytic system and glucose oxidation.
- What about the Randle cycle in cardiomyocytes; has this effect been attributed to defects in glucose-6-phosphatase or is this a defect in glucose transport as shown in skeletal muscle in insulin resistance (J Clin Invest. 2016;126(1):12-22)?
- From what is presented, is the comparison of the Warburg effect documented in cancer sufficient to explain heart failure? Is the concept of aerobic glycolysis seen in cancer really appropriate here? What about the role of pyruvate kinase?
- Figure 1 is nice but would be improved and more consistent with what’s covered in the text if glucose metabolism was included.
- What happens to glucose metabolism in cardiac-specific PPARα-and cardiac-specific ERR-deficient and overexpression mice; and cardiomyocyte-specific KLF4 mice?
- Is heart failure prevented in cardiac-specific PGC-1α overexpressing mice?
Triacylgriceride dynamics and heart failure
- What about the role of lipolysis of cellular triglycerides by adipose triglyceride lipase in generating essential mediator(s) involved in the generation of lipid ligands for PPAR activation (Nature Medicine, Volume17, pages 1076–1085 (2011)?
- What happens to impaired cardiac function when the cardiomyocyte TAG pool is expended?
- What is the mechanism for selective ceramide-induced cardiotoxicity; is this ER stress?
- Some description of the relative roles of de novo fatty acid/TG biosynthesis vs. extracellular uptake, i.e., lipoprotein lipase would be worthwhile.
Membrane fatty acid composition and heart failure
- Fatty acids don’t only distribute in ER membranes but plasma membranes where cellular substrate uptake and hormone action occur.
- Why do changes in membrane fatty acid composition including expression of SCD1 and SIRT1 modify only diastolic and not systolic function?
- Is there any evidence that dietary fatty composition, i.e. more MUFA vs SFA improves myocardial function 0in humans?
Conclusions
- Fibrate-induced hepatic and renal function test abnormalities have had little role flor not pursuing the potential benefit of PPARα induction in humans at risk for or with heart failure.
- Why are ketones mentioned here and not covered before? In man ways this section is not conclusive but introduces new points.
Round 2
Reviewer 1 Report
The author made significant improvement of the review. The revised version can be accepted in the present form.
Author Response
Thanks a lot.
Reviewer 2 Report
Overall, the responses to my previous review are extensive and worthy. A few suggestions remain:
- The revised Section 2.2 needs to be more concise. In addition, the important role of increased glucose diversion via the pentose phosphate shunt pathway in Failing Heart needs to be better represented in Figure 1.
- The inquiry about the role of glucose uptake (GLUT1, GLUT4) including basal and insulin-mediated in heart failure has not been addressed.
- The Warburg effect in cancer cells needs to be brief with relevance to cardiomyocytes in heart failure more concise.
- The modified Section 3.1 is excellent, thank you.
- The beneficial effects of statins on PPAR alpha/gamma in the heart have not been promoted; any more references here?
- "Ketone bodies have been observed not only to exert 423
anti-cardiac effects when utilized as an energy substrate but also to act in an org in an organoprotective manner through signaling [120,121] and epigenetic regulation [122,123]" needs to clarified.
